# Basic Psychological Needs, Burnout and Engagement in Sport: The Mediating Role of Motivation Regulation

**DOI:** 10.3390/ijerph17144941

**Published:** 2020-07-09

**Authors:** Cristina De Francisco, Elisa Isabel Sánchez-Romero, María Del Pilar Vílchez Conesa, Constantino Arce

**Affiliations:** 1Department of Health Sciences, Catholic University of Murcia, 30107 Murcia, Spain; cdefrancisco@ucam.edu; 2Department of Social Sciences and Communication, Catholic University of Murcia, 30107 Murcia, Spain; eisanchez@ucam.edu; 3Department of Social Psychology, Basic and Methodology, University of Santiago de Compostela, 15782 Santiago de Compostela, Spain; constantino.arce@usc.es

**Keywords:** burnout, engagement, basic psychological needs, motivation

## Abstract

The purpose of the present research was to analyze the mediating role of motivational regulation between the satisfaction of basic psychological needs and burnout and engagement in athletes. From different sports 1011 young Spanish athletes participated in the study. Participants completed several measurement instruments concerning: the Basic Needs Satisfaction in Sport Scale, Behavioral Regulation in Sport Questionnaire, Athlete Burnout Questionnaire and Athlete Engagement Questionnaire. The results of structural equation modeling showed that the satisfaction of basic psychological needs has direct effects on burnout and engagement: a negative effect on athlete burnout (–0.49, *p* < 0.001) and a positive effect on engagement (0.54, *p* < 0.001). In addition, the satisfaction of basic psychological needs has a partial indirect effect over these variables in the same direction mediated by the self-determined degree of motivation. Thus, low levels of self-regulated motivation are positively related to burnout, but high levels of self-determined motivation are not. The same was observed with engagement, but vice versa: high levels of self-determined motivation are positively related to athlete engagement, but low levels of self-determined motivation are not. The proposed model explained 37% of the variance of burnout and 51% of the variance of engagement.

## 1. Introduction

Sport burnout is a multidimensional syndrome characterized by the physical and psychological exhaustion derived from the demands of training and competition, which tends to decrease athletes’ performance [1,2]. Self-Determination Theory (SDT) is one of the theoretical frameworks used to explain the origin of athlete burnout [3,4], so studies that relate motivation to sport burnout have increased in recent years [5,6,7].

The regulation of human behavior is one of the central points of SDT. SDT establishes that motivation is composed of six behavioral regulating factors, ranging from higher to lower self-determination: intrinsic regulation, integrated regulation, identified regulation, introjected regulation, external regulation and amotivation [8,9,10]. In this sense, self-determination refers to an innate tendency towards optimal engagement with the environment, such that voluntary and volitional engagement promotes psychological well-being and, in contrast, an interaction controlled by the environment tends to generate personal distress [11]. Thus, the most self-determined degrees of motivation are not dependent on external factors, but are achieved by internal motives [12]. Framed within SDT is the theory of Basic Psychological Needs (BPN) [13]. This theory considers the satisfaction of three psychological needs (competence, autonomy and relatedness) as an important stimulus that governs human behavior [14]. Competence is the belief that one can achieve the desired results. Autonomy is the feeling that one decides one’s actions. Finally, relatedness with others is the perception of belonging to a group. In addition, several authors [15,16] understand autonomy as a variable that depends on three factors: perceived locus of causation, volition and choice. Perceived locus of causation is a two-pole continuum: internal, when people perceive that their behaviors begin and are regulated by personal preferences, and external, when people perceive that their behaviors begin and are regulated by the environment. Volition is the feeling one has when one does what one wishes to do and does not do what one does not wish to do. Choice is the personal ability to decide in flexible or restrictive contexts.

Both theories (SDT and BPN) are linked such that satisfaction of BPN is related to the more self-determined degrees of motivation. There are numerous investigations in the sporting context whose results show that satisfaction of BPN positively predicts the more self-determined levels of motivation (intrinsic motivation, integrated regulation and identified regulation) in football players [17,18]; and negatively predicts the less self-determined levels (introjected regulation, external regulation and amotivation) in different contexts (elite athletes, regional athletes and also in students of Physical Education) [19,20,21,22,23,24]. On another hand, other studies have shown the mediating role of motivation between satisfaction of BPN and other variables such as the degree of well-being experienced [25], satisfaction with life and the development of a positive effect [11].

Various investigations show that satisfaction of BPN has a negative effect on burnout [7,26,27] and a positive effect on engagement [28,29,30], or both [31,32]. For example, a study with Spanish third division soccer players [32] has found positive and negative predictions of the basic psychological needs about engagement and burnout, respectively. The results of a recent review [33] indicated correlations between burnout in elite athletes and motivation or basic psychological needs; more concretely, a negative correlation was found between burnout and intrinsic regulations of motivation and also with basic psychological needs, and positive correlations between burnout and amotivation. In another hand, a sporting career is starting increasingly early. Children and young people spend many hours training at increasingly demanding levels, while they have to study, relate with others and develop towards adulthood. Thus, in many occasions the sports career is a continuum of ups and downs for athletes. Therefore, to know the factors that can condition the sport practice and even the early retirement from a professional career of the athletes. In this sense, under the SDT, the relationships between self-determined motivation toward sport participation and developmental outcomes were examined [34] in youth athletes. Results indicated that participants with higher self-determined motivation toward sport reported higher general self-efficacy more positive attitudes toward a healthy lifestyle, and lower engagement in threatening behavior. In addition, current studies highlight the need to take into consideration a combination of factors to understand the occurrence of burnout in youth athletes [7] and promote early athletic engagement (in youth football players) [35]. In recent years, sport engagement has shown multiple benefits for athletes’ performance and for a positive climate of the sport world, in amateur and elite athletes [36,37,38,39]. Engagement is as a positive and lasting affective-cognitive state that is experienced when a task is carried out, and can vary according to the moment or the task itself. It appears in the labor context as a construct with three dimensions that combine with each other: vigor, dedication and absorption [40]. Various authors establish engagement as the opposite phenomenon of burnout, in some theoretical frameworks [41,42], as people who are engaged feel connected to their work, experiencing it as a challenge and not as a source of stress [43]. Therefore, burnout and engagement correlate inversely, according to a review study [44]. In sport, this idea is maintained, but sport engagement presents a greater conceptual breadth than work engagement because it consists of four interrelated dimensions: vigor, dedication, confidence and enthusiasm, in elite athletes [38,45].

No studies have been found in the literature that have tested models that include at the same time satisfaction of basic psychological needs, regulations of motivation, burnout and engagement. The present research provides important contributions to the understanding of the link among these variables in two senses. First, we test a model that integrates and jointly analyzes the relationships among the satisfaction of basic psychological needs, self-determined motivation, burnout and engagement. Second, we examine the mediating role of self-determined motivation in the relationship between the satisfaction of basic psychological needs and burnout, and between the satisfaction of basic psychological needs and engagement. In addition to these two contributions, we check if the model was invariant in relation to gender in order to know if it can be applied interchangeably for men and women. Therefore, the present study aims to analyze the relationship between BPN and burnout and engagement, taking into account the mediating role of motivation, as a first step in the study of these variables. Figure 1 provides a schema of the main study hypotheses:

Satisfaction of BPN has a direct negative effect on burnout and a positive one on engagement, as these two concepts are inversely related to each other.

The effect of satisfaction of BPN on burnout and engagement is mediated by the athlete’s degree of self-determined motivation, with BPN satisfaction maintaining:(a)An inverse relationship with lower levels of self-determined motivation, which in turn has a positive effect on burnout and a negative effect on engagement and;(b)A positive relationship with the higher levels of self-determined motivation, which, in turn, has a negative effect on burnout and a positive effect on engagement.

## 2. Materials and Methods

We selected 1011 Spanish athletes of multiple sport modalities through an intentional non-probabilistic sampling. The sample was gender-balanced (505 males and 506 females) with a mean age of 18.09 years (*SD* = 5.55). The collective sport modality (71.4%) predominated over the individual modality, with soccer being the most practiced sport (44.4%), followed by athletics (9.5%) basketball (7.3%), futsal (6.4%), handball (4.6%), rugby (3.6%), volleyball and taekwondo (3.3% each one), rhythmic gymnastics (3.1%) and other sports (*n* = 18 with 1% or less: 14.5%). Of the athletes 11.9% had a low competitive profile (local categories), 63.2% had a medium competitive profile (regional) and 24.3% had a high level (international and national categories). The mean total time in sports practice was 8.07 (*SD* = 4.68). Participants trained on average more than three sessions per week (*M* = 3.38, *SD* = 3.08), with an average duration of 101.23 min per session (*SD* = 38.17), for approximately 10 months per year (*M* = 9.95, *SD* = 1.27).

The Basic Needs Satisfaction in Sport Scale (BNSSS) [23] measures the satisfaction of five basic psychological needs proposed by its authors through 20 items: competence (five items), autonomy_choice (four items), autonomy_volition (three items), autonomy_Perceived Locus of Internal Causation (IPLOC, three items) and relatedness (five items). Each item is rated on a Likert-type response format ranging from 1 (not at all true) to 7 (completely true). We used the Spanish version [46], which presents adequate psychometric properties of validity (χ^2^/df = 3.12; RMSEA = 0.06, NNFI = 0.96 and CFI = 0.97) and composite reliability (all values were higher than 0.70).

The Behavioral Regulation in Sport Questionnaire (BRSQ) [47] evaluates six degrees of motivation regulation: amotivation (without regulation), external regulation, introjected regulation, identified regulation, integrated regulation and intrinsic motivation. The Likert-type response format ranges from 1 (completely false) to 7 (completely true). The 24-item Spanish version [10] equally distributed for all six degrees, was applied. The authors reported a good fit of the model through confirmatory factorial analysis (χ^2^/df = 3.44, RMSEA = 0.07, CFI = 0.92 and TLI = 0.91) and reliability (the six degrees show values higher than 0.70). In order to facilitate the interpretation of the model, it was decided to regroup the six dimensions into the following two categories as in previous research [48]: high self-motivation, with the three most regulated dimensions (intrinsic regulation, integrated regulation and identified regulation) and low self-motivation, with three least regulated dimensions (amotivation, external regulation and introjected regulation). Previous studies within the SDT [49] have grouped the different levels of regulation into two levels (autonomous vs. controlled) with the same intention of simplifying the model.

The Athlete Burnout Questionnaire (ABQ) [1,50] measures athletes’ burnout syndrome and contains the dimensions: physical/emotional exhaustion, devaluation and reduced sense of achievement. The Likert-type response format ranges from 1 (almost never) to 5 (almost always). We used the 9-item Spanish version (three items per dimension) [51], which has shown to be a valid (χ^2^/df = 1.78, RMSEA = 0.04, NFI = 0.95, TLI = 0.96 and CFI = 0.98) and reliable instrument (values of Cronbach alpha between 0.64 and 0.80).

The Athlete Engagement Questionnaire (AEQ) [45] measures engagement in athletes through four dimensions: confidence, vigor, dedication and enthusiasm. The 16-item (four items for each dimension) Spanish version of AEQ was administered [52]. It uses a Likert-type response format ranging from 1 (almost never) to 5 (almost always), like the original version. The Spanish version has adequate psychometric properties regarding its factorial validity (χ^2^/df = 3.12, RMSEA = 0.06, CFI = 0.95, GFI = 0.93, TLI = 0.94 and SRMR = 0.054) and reliability (values of Cronbach alpha higher than 0.80). In a subsequent study, its construct validity was also demonstrated [53].

We contacted the coaches and/or sports directors of selected clubs to explain the purpose of the study and to schedule the days of data collection with the athletes. Questionnaires were completed on paper in the presence of one research assistant. Prior to the administration of the questionnaire, athletes (sports tutors in the case of minors) signed the data transfer sheet approved by the ethics commission of the university.

A retrospective ex post facto design was carried out with a single group and multiple measurements, as the sample was selected as a function of the greatest possible heterogeneity in the variables that explain or moderate engagement and burnout [54].

We started with an exploratory data analysis in search of missing values and outliers. Then, an initial description of the data was made with basic descriptive statistics and the relationship between variables was verified with Spearman correlations through the IBM SPSS version 24, in view of the non-normality of the variables. Reliability was measured in terms of internal consistency using the cutoff value of α > 0.70 [55]. Next, a structural equation model was carried out with IBM AMOS, version 22 [56], in which the relationships were established between satisfaction of the basic psychological needs measured through the BNSSS, motivational regulation measured with the BRSQ and engagement and burnout, measured through the total scores of the AEQ and the ABQ, respectively. The Mardia test was used to verify the multivariate normal distribution, maximum likelihood to estimate the parameters of the model and bootstrap to estimate their standard errors and confidence intervals. In order to assess the overall fit of the model were used the χ^2^ statistic, the Tucker–Lewis Index (TLI), the Comparative Fit Index (CFI), the Root Mean Square Error of Approximation (RMSEA) and the Standardized Root Mean Square Residual (SRMR). For these indexes, the following cutoff values were adopted: TLI and CFI ≥ 0.90 and RMSEA and SRMR ≤ 0.08 [57,58,59]. Lastly, the invariance of the model was tested evaluating the statistical significance of the change in χ^2^ (Δ χ^2^) between nested models and the change in CFI (ΔCFI), considering differences of less than 0.01 non relevant [60].

## 3. Results

### 3.1. Exploratory Data Analysis

Very few items contained missing values and in all cases the percentage was less than 1%. The missing values for item *i* were replaced by the most frequent response of the subject *j* to the items of the factor to which item *i* belonged. No outliers were observed.

### 3.2. Descriptive Statistics and Correlations

Table 1 shows the means and standard deviations of the variables of the study. Regarding the satisfaction of BPN, measured on a scale of 1–7, the satisfaction of the need of autonomy_volition presented the highest mean (*M* = 6.22), whereas the lowest mean referred to the satisfaction of the need of autonomy_choice (*M* = 5.07). With respect to motivation, also measured on a scale of 1–7, the least self-determined degrees (amotivation, external regulation and introjected regulation) presented a lower mean (*M* = 2.06) than the more self-determined levels (intrinsic regulation, integrated regulation and identified regulation; *M* = 5.90).

It can also be observed in Table 1 that all the Spearman correlations between factors had statistically significant values, with the exception of the relationship between low self-determined motivation and high self-determined motivation (*ρ* = −0.06, *p* > 0.05). With respect to the sign of values, the positive signs predominated. The negative values refer to the following relationships: satisfaction of all BPN with low self-determined motivation; satisfaction of all BPN with burnout; high self-determined motivation with burnout; low self-determined motivation with high self-determined motivation; low self-determined motivation with engagement; and burnout with engagement. The strongest relationship was observed between burnout and engagement (*ρ* = −0.62, *p* < 0.01), and the weakest relationship observed was between autonomy_choice and low self-determined motivation (*ρ* = –0.08, *p* < 0.05). Lastly, Table 1 shows the values of the Cronbach alpha coefficients for each of the studied factors on the main diagonal of the matrix. All values were above the threshold of 0.70, with the exception of autonomy_volition, which was below it (0.53).

### 3.3. SEM Model

In consequence with the hypothesis formulated, a model was specified with satisfaction of BPN as an exogenous variable (a general factor with five indicators corresponding to each one of the primary factors of the BNSSS scale), two endogenous variables (burnout and engagement), allowing the correlation between their error terms and two mediator variables (low and high self-determined motivation). The specified model contained 45 different sample moments, 23 parameters to be estimated and 22 degrees of freedom (Figure 2).

The model explained 37% of the variance of burnout (R^2^ = 0.37) and 51% of the variance of engagement (R^2^ = 0.51). In addition, satisfaction of the BPN explained 6% of the variance of low self-determined motivation (R^2^ = 0.06) and 53% of the variance of high self-determined motivation (R^2^ = 0.53). The global fit indexes were χ^2^(22) = 262.300 (*p* < 0.001), TLI = 0.90, CFI = 0.94, RMSEA = 0.10 (90%), CI [0.09, 0.11] and SRMR = 0.05.

All parameter estimates are in the expected direction (positive, negative), and are also statistically significant (*p* < 0.01) except for the effect of low self-determined motivation on engagement (−0.05, *p* = 0.08) and the effect of high self-determined motivation on burnout (−0.03, *p* = 0.34). BPN satisfaction had an indirect effect mediated by (high and low) self-motivation over burnout of −0.08 (95% CI −0.008, −0.192) and over engagement of 0.17 (95% CI 0.097, 0.27).

### 3.4. Invariance of the Model across Gender

Table 2 offers the values of χ^2^ and CFI for four different measurement models: unconstrained, measurement weights, structural covariances and measurement residuals. The difference in χ^2^ between the measurement weights and unconstrained models did not reach statistical significance (χ^2^_dif(8)_ = 12.817; *p* > 0.05); nor did the differences between the structural covariances and unconstrained models (χ^2^_dif(13)_ = 18.78; *p* > 0.05). Additionally, on the contrary, the difference between measurement residuals and unconstrained models reached statistical significance (χ^2^_dif(23)_ = 67.533; *p* < 0.05). The lack of statistical significance, in the first difference analyzed, can take as evidence of homogeneous factor loadings in both groups of athletes (male and female); while the second χ^2^_dif_ value indicated there was homogeneity in the correlations and variances of the factors in both groups; and the statistical significance of the third difference, evidenced the lack of homogeneity of variance of the errors and their correlations in both groups. These findings were confirmed by the observation that all values of CFI_dif_ between models were less than 0.01, with the exception of the difference between measurement residuals and unconstrained models that was slightly above (0.011).

## 4. Discussion

The objective of the present study was to analyze the relationship between the satisfaction of BPN, burnout and engagement, and to describe the mediating role of motivation. To this end, a model of athlete burnout based on SDT was tested. This theory is one of the theories with the most predictive power to explain burnout [61].

### 4.1. Relationship between Satisfaction of BPN and Burnout and Engagement

The first hypothesis, which proposed a negative effect of satisfaction of BPN on burnout, was confirmed. Similar data were indicated by previous investigations in Canadian young adults [29] and in English junior athletes [31], which showed that satisfaction of the BPN had a direct negative effect on burnout in young athletes of both sexes of different sports modalities, as in the sample of this work. This effect could be explained by the nature of the BPN, for example, when athletes feel more competent, they firmly believe they can achieve good results, an attitude contrary to burnout, in which athletes do not feel capable of achieving their goals because of a very much reduced sense of achievement [62]. Additionally, a significant negative impact on the perceptions of competence can influence the value that athletes assign to their sport [63]. When they feel less autonomous, the lack of psychological freedom could bring about them to run out of external resources causing an imbalance between demands and resources. It also makes individuals feel insecure about achieving high performance (reduced personal achievement) and the tendency to be negative about work [64]. Finally, as the need for relatedness can be considered as the feeling of belonging to the social environment [64], it is logical to think that if this need is not satisfied, it creates a distance from the practice and the sports agents involved (devaluation of sports practice dimension).

Regarding the hypothesized relationship between the satisfaction of BPN and engagement, the results showed that, effectively, satisfaction of BPN had a positive effect, like another previous study [31], which showed that the satisfaction of BPN explained 71% of the variance of engagement in young English athletes. Satisfaction of the needs (particularly competence and autonomy) was a good predictor of engagement in Canadian athletes, explaining 30% of the variance [29]. This relationship may be due to athletes’ behavioral implication to satisfy their BNP, which in turn promotes an increase of engagement [28].

### 4.2. Mediated Role of Degree of Regulated Motivation between Satisfaction of BPN and Burnout and Engagement

The results showed that satisfaction of BPN had direct effects on self-determined motivation, in the sense that it is positively related to high self-determined motivation, and negatively related to low self-determined motivation. The results of other investigation in Spanish athletes from multiple sports [48] estimate the percentage of explained variance between 50% and 6%, similar to those found in this research (53% and 6%). Similarly, satisfaction of BPN explained 38.8% of the variance of the levels of more self-determined motivation and 5.7% of the less self-determined levels, in cross-fit participants [49]. It therefore coincides with previous studies that conclude that satisfaction of BPN is related to a higher percentage of explained variance of high self-determined motivation with respect to low self-determined motivation. It is likely that athletes who satisfy BPN feel motivated through more internal regulations, such as interest in the activity itself or the connection between sports practice and other personal goals, in youth sport [28]. This could be because, according to SDT, people will decide to carry out activities to satisfy BPN and, once satisfied; they feel more intrinsically motivated to participate in other activities [60]. However, when athletes do not enjoy the sporting practice and do it for other reasons, they do not feel competent, autonomous or related, but instead they practice because they feel obliged, or to avoid feeling guilty if they do not practice, for example, which is when they feel less self-determined or feel more extrinsic regulations.

According to the second hypothesis, the effect of satisfaction of BPN on burnout and engagement would be mediated by the athlete’s degree of self-determined motivation, with the satisfaction of BPN maintaining two types of relations: a negative relation with the lowest levels of self-determined motivation, which, in turn, have a positive effect on burnout and a negative effect on engagement; satisfaction of BPN also has a positive relationship with the highest levels of self-determined motivation, which, in turn, have a negative effect on burnout and a positive one on engagement. Although this hypothesis is consistent with previous studies in the literature [3], showing the usefulness of the self-determination theory in the explanation of burnout, it was partially confirmed by the data of this investigation, as no statistical significance was found for the effects of high self-determined motivation on burnout or of low self-determined motivation on engagement. Perhaps when athletes feel very high self-motivation, other variables are more important to athletes’ perception or the chances of preventing burnout, like passion [65]. It is also possible that motivation regulations are not isolated variables but form a continuum. Regulations that are considered intrinsic in the literature are halfway between intrinsic motivation itself and extrinsic regulations, and these regulations could weaken the prevention of less self-determined levels of motivation against burnout.

In reference to the relationship between self-determined motivation and burnout, the results showed a positive and moderate relationship between the less self-determined levels of motivation and burnout. Less self-determined motivation levels were positively related to burnout, in youth football coaches [4]. As in the present research, these authors found moderate positive correlations between the less self-determined degrees of motivation and burnout, but through its three dimensions, reduced sense of achievement (*r* = 0.56), emotional exhaustion (*r* = 0.31) and sports devaluation (*r* = 0.61). More recently, a positive and significant relationship between amotivation and burnout also was observed, in youth athletes [7]. Analyzing the different degrees of regulation of self-determined motivation, found that amotivation was positively and significantly related to the three components of burnout separately (emotional exhaustion, reduced feeling of achievement and sports devaluation), in elite athletes [66]. Motivation requires energy, direction and persistence, whereas amotivation reflects the opposite, lethargy, apathy and indifference, commonly used terms to describe the symptoms of burnout in general, and more specifically, of the devaluation of sports practice, in youth athletes [61].

Finally, the results about the relationship between self-determined motivation and engagement showed that the most self-determined levels had a positive effect on sport engagement, explaining 21% of its variance. Very similar results were observed in other research [30], where 15% of the variance of engagement was explained by intrinsic motivation. In this research, the less self-determined levels of motivation did not have a significant effect on engagement. When athletes experience amotivation, they find no incentive for sports practice, and therefore do not intend to continue practicing (so there would be no engagement), in youth water polo players [30].

Satisfaction of BPN in Spanish athletes had a direct (positive) effect on engagement and an indirect effect mediated by motivation, explaining 41% of the variance of the model, whereas in the present study, we obtained 51% [48]. This model with four variables that examined the effect of satisfaction of BPN on sport burnout and engagement, with self-determined motivation acting as a mediator variable, has not been studied frequently in the sports field. We found a single study in youth athletes [67] that analyzes the relationship between the variables measured in this article (satisfaction of BPN, self-determined motivation, sport burnout and engagement), but without investigating the mediator role of motivation. The authors established different profiles of sport sense of community (which included a dimension to measure the satisfaction of needs) according to the levels of shown in motivation, sport burnout and engagement); through a multiple regression, they examined the effect of the sport sense of community on motivation, burnout and engagement at two temporal moments. The results revealed sport sense of community as a protective factor against burnout, and as a factor promoting engagement and self-determined motivation.

This research is so novel because it provides data on the explanatory power of a model that combines satisfaction of basic psychological needs, regulation of motivation, burnout and engagement. Previous research have shown the relationships among satisfaction of basic psychological needs, self determined motivation, burnout and engagement, but not in a single model, for example: (1) satisfaction of BPN predicts positively high levels of self-determined motivation [17,18] and negatively low self-determined motivation levels [19,20,21,22,23,24]; (2) satisfaction of BPN has a negative effect on burnout and positive on engagement [32], (3) motivation has a mediating role between satisfaction of BPN and engagement [48] and burnout [27]. Regarding the joint relationship of all variables, until now it has only shown that athletes with a high sporting sense of community profile (which included the satisfaction of BPN), report lower burnout scores, have higher engagement scores and are characterized by self-determined regulatory styles [66], but we did not have verified data for this model.

### 4.3. Practical Applications

Reducing the degree of burnout experienced by athletes and promoting sport engagement is essential to achieve people’s sport motivation and commitment to physical activity [18]. Although research on predictors and mediators of sport engagement is still in its initial stages, it has many implications for athletes and professionals in the sports sector, due to its relevance for performance, competition, health and physical endurance [68]. According to the findings of this study, strategies to prevent burnout and promote sport engagement should be focused on supporting satisfaction of BPN and encouraging high self-determined motivation in athletes, while the less self-determined levels are not so important. For example, it is important to remember for athletes why they practice sports and what they like most, reinforcing intrinsic motivation of the athlete. In this sense, identifying the reasons for feeling motivated to carry out physical sport activity is an essential aspect for the development and implementation of effective sports interventions and in accordance with real needs. In addition, sports programs should pay attention to parents and coaches, and work to ensure that athletes receive adequate support [69].

### 4.4. Limitations and Future Research

This study also has some limitations. First of all, we need to point out that although the goodness of fit indexes of the model were in general satisfactory, the value of RMSEA of 0.10 was slightly above recommendations (0.08). Secondly, we must take into account there are many different sports, with a totally different nature, where differences could be found between the way of thinking and feeling the sport of each athlete. It could be interesting to analyze how each athlete satisfies BPN, and what predictive power have motivation over burnout, for example, in individual and collective athletes separately, or in strength sports such as judo or weightlifting versus endurance sports such as triathlon or athletics, on the other hand.

Finally, the analysis of the mediating role of motivation in the relationship of BPN with sport burnout and engagement is not an area that has been studied very frequently. It would be interesting to continue investigating the relationships of these variables, due to the positive consequences that engagement can provide in athletes and, on the contrary, the negative consequences that athletes present when they show high degrees of burnout. This work tries to be a first step to establish a theoretical model in the sports field based on the SDT. However, it is necessary to continue deepening this model in relation to the role of each BPN and each degree of motivational regulation on burnout and engagement. In addition, despite the fact that the relationship between satisfaction of the BPN and athletes’ engagement is demonstrated, research in Spanish-speaking athletes has not received appropriate attention, due to the, until recently, non-existence of a measuring instrument of sport engagement validated for this population [52].

## 5. Conclusions

In this research, the satisfaction of BPN exerted a direct effect on sport burnout and engagement, and an indirect effect through two mediating variables: low and high self-determined motivation. For practical purposes, the satisfaction of BPN was simplified as a latent global factor, and the levels of motivation regulation were dichotomized (high and low self-determined motivation). The results support the model in relation to the mediating role of the levels of motivational regulation, albeit partially, as no relationship was observed between high self-determined motivation and burnout, or between low self-determined motivation and engagement. In addition, this model was tested across gender, finding evidence of its invariance in the unconstrained model, in measurement weights and in structural covariances.

## Figures and Tables

**Figure 1 ijerph-17-04941-f001:**
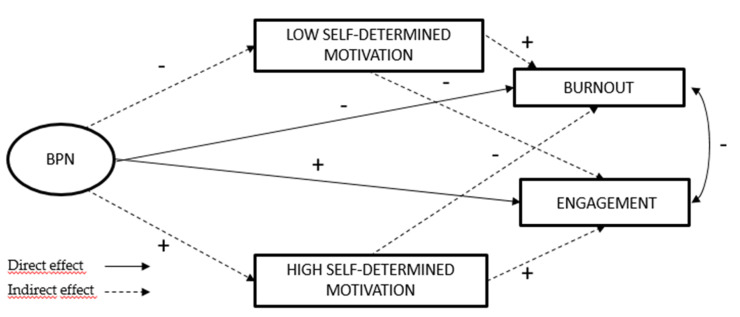
Hypothetical model among satisfaction of Basics Psychological Needs (BPN), self-determined motivation, burnout and engagement.

**Figure 2 ijerph-17-04941-f002:**
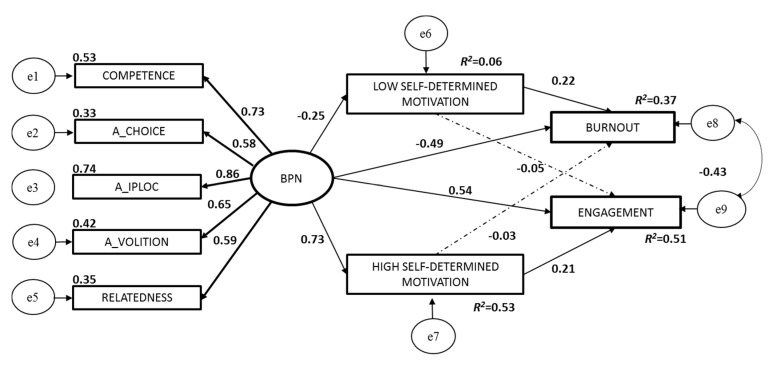
Structural model of satisfaction of Basics Psychological Needs (BPN), self-determined motivation, burnout and engagement (standardized path, error coefficients -e- and explained variance -R^2^-). Note: 
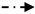
 not statistically significant.

**Table 1 ijerph-17-04941-t001:** Descriptive statistics and correlation matrix for the satisfaction of basic psychological needs, motivation, burnout and engagement.

7.	*M* (*SD*)	1	2	3	4	5	6	7	8	9
1. Competence	5.69 (1.00)	*0.87*	0.51 **	0.39 **	0.60 **	0.42 **	−0.11 **	0.53 **	−0.42 **	0.59 **
2. Autonomy_ choice	5.07 (1.27)		*0.81*	0.31 **	0.49 **	0.37 **	−0.08 *	0.40 **	−0.22 **	0.33 **
3. Autonomy_ volition	6.22 (0.89)			*0.53*	0.57 **	0.36 **	–0.38 **	0.45 **	–0.44 **	0.47 **
4. Autonomy_ IPLOC	5.89 (1.06)				*0.74*	0.44 **	–0.26 **	0.59 **	−0.48 **	0.60 **
5. Relatedness	6.13 (0.90)					*0.79*	−0.17 **	0.47 **	−0.29 **	0.33 **
6. Low self-determined motivation	2.06 (1.10)						*0.89*	−0.06	0.34 **	0.18 **
7. High self-determined motivation	5.90 (0.84)							*0.86*	−0.35 **	0.57 **
8. Burnout	1.98 (0.57)								*0.74*	−0.62 **
9. Engagement	4.19 (0.58)									*0.92*

Cronbach’s alpha values on the diagonal; ** *p* < 0.01 * *p* < 0.05.

**Table 2 ijerph-17-04941-t002:** Measurement models for the study of invariance by gender.

Model	χ^2^	df	*p*-Value	CFI
Unconstrained	300.98	44	<0.001	0.93
Measurement weights	313.80	52	<0.001	0.93
Structural covariances	319.76	57	<0.001	0.93
Measurement residuals	368.53	67	<0.001	0.92

Note: df = Degrees of Freedom, CFI = Comparative Fit Index.

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
