# Peer review of "Basic Psychological Needs, Burnout and Engagement in Sport: The Mediating Role of Motivation Regulation"

_ijerph, 2020, doi:10.3390/ijerph17144941_

Round 1

Reviewer 1 Report

I think the current study proves its value through well explained and justified regression models, even though the fit index is slightly above of recommendations. Nevertheless, I have some concerns, first I will acknowledge specific concerns throughout the paper, and afterwards general and even more relevant concerns of the research.

  1. Introduction

2nd paragraph, modify writing is not clear “In addition, various authors [15,

16] defend divide autonomy into three components: perceived locus of causation, volition, and choice.”

4th paragrah, please clrarify the direction of the cited correlations “The results of a recent review indicated correlations between burnout in elite athletes and motivation, and between burnout and basic psychological needs, but it is necessary to know the way they affect each other.”

  1. Materials and Methods

Specify average of total time in sports practice, there was criteria for selection in this regard? Is not the same to have been practicing for ten years rather than 6 months… Further on authors report about the application of a “booklet” if there is relevant data to report please include a table, otherwise this paragraph is just anecdotic and does not help the paper.

  1. Discussion

Mediated role of degree of regulated motivation between satisfaction of BPN and burnout and engagement

I think that authors make and strong effort to describe their own results on relation with previous studies, when results replicate themselves the interpretation is easy, even the citate others works in this regard, nevertheless when results are not as expected this section lacks a deeper theoretical explanation of the possible reasons.

General remarks

As a reader, I failed to find a good justification of the contributions of the current paper, authors explained very well the measured variables, and constantly trough the introduction and the discussion clarify that this questions have been answered before, in this regard they direct their hypothesis in a very logic way with support of previous findings, nevertheless I think the current state of the paper doesn´t convince the reader that this research design contribute new findings, maybe is the writing, I encourage authors to improve the introduction and discussion sections in order the clearly remark (in short) what does this research contributes that others don´t. What is the novelty about your study?

I understand that these kinds of models try to predict variables behaviors trough regression models and the data cannot be treated as causal, however I think that you have a robust amount of data to prove a different (and maybe more relevant approach). For example, I´m worried that given the nature of your sample (professional athletes / aspirants to be professionals / or at least people with a clear habit to sport practice) the distribution of your sample could be biased towards purely “positive” attributes of sport perception, good motivation and positive self-determination related variables. Wouldn´t be more interesting and theoretical enriching to clearly identify those athletes that are in the right direction of their perceived motivation to sport practice (as other studies already found) and those who practice sport but for the “wrong” or “maladaptive” reasons?

In this regard and other statistical approach, you could land in more novel conclusions.

I hope authors can attend these concerns, or at least provide a convincing response of the high value of the current paper.

Author Response

Dear reviewers and editor, the changes made in the text are explained below according to the suggestions of each reviewer. We appreciate your comments as we improve the quality of our work.

Revisor 1

  1. Introduction

 2nd paragraph, modify writing is not clear “In addition, various authors [15, 16] defend divide autonomy into three components: perceived locus of causation, volition, and choice.”

 Done, it has been explained much better now. Thanks for the comment.

4th paragraph, please clarify the direction of the cited correlations “The results of a recent review indicated correlations between burnout in elite athletes and motivation, and between burnout and basic psychological needs, but it is necessary to know the way they affect each other.”

Yes, indeed, it has been rewritten to indicate which relationship and direction were between the variables according to this important review study. Thanks for this useful comments.

  1. Materials and Methods

Specify average of total time in sports practice, there was criteria for selection in this regard? Is not the same to have been practicing for ten years rather than 6 months… 

Done. We have specified the range and have expanded the description of the participants, responding to another comment of other revisor: “The highest percentage of participation in terms of modality of sport was in soccer (44.4%%), followed by athletics (9.5%) basketball (7.3%), futsal (6.4%), handball (4.6%), rugby (3.6%),volleyball and taekwondo (3.3% each one), rhythmic gymnastics (3.1%) and other sports (n = 18 with 1% or less: 14.5%). 11.9% of the athletes had a low competitive profile (local categories), 63.2% had a medium competitive profile (regional), and 24.3% had a high level (international and national categories). The mean total time in sports practice was 8.07 (SD = 4.68)”.

Further on authors report about the application of a “booklet” if there is relevant data to report please include a table, otherwise this paragraph is just anecdotic and does not help the paper.

We appreciate that the reviewer has commented on this detail. Since the reviewer considers it  anecdotic, we removed it this paragraph:: A booklet was developed in which we initially asked about demographic (age, sex) and sports data (sport practiced, number of weekly trainings and duration and months of practice). The Spanish versions of the questionnaires were then presented for the measurement of the study variables. 

  1. Discussion

 Mediated role of degree of regulated motivation between satisfaction of BPN and burnout and engagement

 I think that authors make and strong effort to describe their own results on relation with previous studies, when results replicate themselves the interpretation is easy, even the citate others works in this regard, nevertheless when results are not as expected this section lacks a deeper theoretical explanation of the possible reasons.

We have tried to respond to what is requested. When the relationships are not very strong, perhaps other variables should be considered to explain some assumptions. We have done with some studies that are agreed. 

  1. General remarks

As a reader, I failed to find a good justification of the contributions of the current paper, authors explained very well the measured variables, and constantly trough the introduction and the discussion clarify that this questions have been answered before, in this regard they direct their hypothesis in a very logic way with support of previous findings, nevertheless I think the current state of the paper doesn´t convince the reader that this research design contribute new findings, maybe is the writing, I encourage authors to improve the introduction and discussion sections in order the clearly remark (in short) what does this research contributes that others don´t. What is the novelty about your study?

We extended introduction and discussion sections in order to highlight of novelty of our study

Introduction: 

No studies have been found in the literature that have tested models that include at the same time satisfaction of basic psychological needs, regulation of motivation, burnout and engagement. The present research provides important contributions to the understanding of the link among these variable in two senses. First, we test a model that integrates and jointly analyzes the relationships among satisfaction of basic psychological needs, self determined motivation, burnout and engagement. Second, we examine the mediating role of self-determined motivation in the relationship between satisfaction of basic psychological needs and burnout, and between satisfaction of basic psychological needs and engagement. In addition to these two contributions, we check if the model was invariant in relation to gender in order to know if this model can be applied interchangeably for men and women.

Discussion:

This research is so novel because it provides data on the explanatory power of a model that combines  satisfaction of basic psychological needs, regulation of motivation, burnout and engagement. Previous researches have shown the relationships among satisfaction of basic psychological needs, self determined motivation, burnout and engagement, but not in a single model, for example: 1) satisfaction of BPN predicts positively high levels of self-determined motivation [17, 18] and negatively low self-determined motivation levels [19-24]; 2) satisfaction of BPN has a negative effect on burnout and positive on engagement [3, 32], 3) motivation has a mediating role between satisfaction of BPN and engagement [48] and burnout [27]. Regarding the joint relationship of all variables, until now it has only shown that athletes with a high sporting sense of community profile (which included the satisfaction of BPN), report lower burnout scores, have higher engagement scores, and are characterized by self-determined regulatory styles [67], but we did not have verified data for this model.

I understand that these kinds of models try to predict variables behaviors trough regression models and the data cannot be treated as causal, however I think that you have a robust amount of data to prove a different (and maybe more relevant approach). For example, I´m worried that given the nature of your sample (professional athletes / aspirants to be professionals / or at least people with a clear habit to sport practice) the distribution of your sample could be biased towards purely “positive” attributes of sport perception, good motivation and positive self-determination related variables. Wouldn´t be more interesting and theoretical enriching to clearly identify those athletes that are in the right direction of their perceived motivation to sport practice (as other studies already found) and those who practice sport but for the “wrong” or “maladaptive” reasons?

In this regard and other statistical approach, you could land in more novel conclusions.

I hope authors can attend these concerns, or at least provide a convincing response of the high value of the current paper.

We totally agree with the reviewer and find the proposal absolutely interesting but unfortunately it was not among the objectives of the study and address it would imply so many changes we cannot afford in the short-term.

Reviewer 2 Report

Thanks for your effort for working on this paper.

The significance of this research is that it provides a basis for athlete’s satisfaction with BPN had a direct effect on burnout and engagement, and indirect influence of motivation regulation in athletes.

The topic, introduction, research design, results, discussion and conclusion of the study is relevant for publication in IJERPH.

Please present the significance of this study more in detail.

Modify the old references to the latest literature.

Reviewer 3 Report

This study explores how motivation influences the relationship between satisfying basic psychological needs and (1) athlete burnout and (2) engagement.

Strengths include: large sample size, use of validated survey tools (questionnaires), focus on a Spanish population that has historically been overlooked

Areas for improvements:

  • More descriptive data about the participants is needed. For example, what level were the athletes (community/recreational, elite, etc)? How many clubs participated? What types of sports (e.g., __% soccer, __ % handball)? When was the data collected?
  • How were the questionnaires completed? On paper? Online? With the assistance of researcher(s)? 
  • Please describe the process of gaining consent from participants. What protocols were used for participants who were not of legal age?
  • Figure 1. Is it possible to use labeling/formatting to distinguish direct, indirect, and mediating effects?
  • Table 1. Reformat so that numbers and stars are not broken between lines. Differentiate (e.g., italicize) the text for Cronbach's alpha. Provide a more detailed note/legend to allow the table to stand alone.
  • Page 5: paragraph below table. Please double-check the strongest and weakest relationships. I believe strongest = burnout and engagement (p=-0.62), autonomy choice = low self-determined motivation (p=-0.08). Note: the relationship between low- and high-self determined motivation is non-significant; there is no relationship.
  • Figure 2: Provide more detailed labeling and legend to allow the figure to stand alone. For example, what e__ represent? Write out abbreviations. Explain that .__ represents r2
  • Discussion/Introduction: Provide more context about past studies (e.g., types/levels of athletes, age, location) and how they fit with the current investigation. 

Round 2

Reviewer 3 Report

The authors have significantly improved this manuscript based on reviewer feedback. Minor editing is required to correct typos and English language errors:

  • Page 7: "All parameter estimates are of in expected direction" should be "All parameter estimates are in the expected direction" 
  • Page 8: "like other preview study" should be "like another previous study"
  • Page 10 "we need to point although the goodness of fit" should be "we need to point out that although the goodness of fit"

Author Response

Thank you very much for reviewing the mistakes, we really appreciate your comprehensiveness.
